# Evaluation of Two Real-Time, TaqMan Reverse Transcription-PCR Assays for Detection of Rabies Virus in Circulating Variants from Argentina: Influence of Sequence Variation

**DOI:** 10.3390/v13010023

**Published:** 2020-12-25

**Authors:** Diego A. Caraballo, María A. Lombardo, Paula Becker, María S. Sabio, Cristina Lema, Leila M. Martínez, Fernando J. Beltrán, Yu Li, Daniel M. Cisterna

**Affiliations:** 1Instituto de Zoonosis “Luis Pasteur”, Av. Díaz Vélez 4821, Ciudad Autónoma de Buenos Aires C1405DCD, Argentina; mariale.lombardo@gmail.com (M.A.L.); paulabecker82@gmail.com (P.B.); ferbelt@hotmail.com (F.J.B.); 2Servicio de Neurovirosis, Instituto Nacional de Enfermedades Infecciosas, Administración Nacional de Laboratorios e Institutos de Salud (ANLIS), “Dr. Carlos G. Malbrán”, Av. Vélez Sarsfield 563, Ciudad Autónoma de Buenos Aires C1282AFF, Argentina; ssoledad@anlis.gob.ar (M.S.S.); clema@anlis.gob.ar (C.L.); lei.mmartinez@gmail.com (L.M.M.); dancis99@yahoo.com (D.M.C.); 3Poxvirus and Rabies Branch, Division of High Consequence Pathogens and Pathology, National Center for Emerging and Zoonotic Infectious Diseases, Centers for Disease Control and Prevention, 1600 Clifton Road, Atlanta, GA 30329, USA; yuli@cdc.gov

**Keywords:** rabies virus, real time RT-PCR, nucleoprotein gene, phylogeny, diagnostic

## Abstract

In rabies diagnosis, it is essential to count on a rapid test to give a quick response. The combined sensitivity and robustness of the TaqMan RT-PCR assays (qRT-PCR) have made these methods a valuable alternative for rabies virus (RABV) detection. We conducted a study to compare the applicability of two widely used qRT-PCR assays targeting the nucleoprotein gene (LysGT1 assay) and leader sequences (LN34 qRT-PCR assay) of RABV genomes, in all variants circulating in Argentina. A total of 44 samples obtained from bats, dogs, cattle, and horses, that were previously tested for rabies by FAT and conventional RT-PCR, were used in the study. All variants were successfully detected by the pan-lyssavirus LN34 qRT-PCR assay. The LysGT1 assay failed to detect three bat-related variants. We further sequenced the region targeted by LysGT1 and demonstrated that the presence of three or more mismatches with respect to the primers and probe sequences precludes viral detection. We conclude that the LysGT1 assay is prone to yield variant-dependent false-negative test results, and in consequence, the LN34 assay would ensure more effective detection of RABV in Argentina.

## 1. Introduction

The rabies virus (RABV), the prototype of the *Lyssavirus* genus, family *Rhabdoviridae*, is the etiological agent of a fatal infection of the nervous system of mammals [1]. The genus has been initially divided into seven genotypes based on the nucleotide sequence of the nucleoprotein gene [2,3] and then confirmed by sequences of the glycoprotein [4] and the phosphoprotein [5] genes. Currently, according to the International Committee on Taxonomy of Viruses (ICTV), the genus *Lyssavirus* has been subdivided into 17 species, based on genetic distance, immunological features, and antigenic patterns in reactions with panels of antinucleocapsid monoclonal antibodies [1]. This classification is also supported by geographic distribution and host range. The availability of complete genome sequences made possible the identification of three distinct lyssavirus phylogroups: phylogroup I comprises RABV, *Aravan lyssavirus* (ARAV), *Australian bat lyssavirus* (ABLV), *Bokeloh bat lyssavirus* (BBLV), *Duvenhage lyssavirus* (DUVV), *European bat 1 lyssavirus* (EBLV-1), *European bat 2 lyssavirus* (EBLV-2), *Gannoruwa bat lyssavirus* (GBLV), *Irkut lyssavirus* (IRKV) and *Khujand lyssavirus* (KHUV); phylogroup II includes the *Lagos bat lyssavirus* (LBV), *Mokola lyssavirus* (MOKV) and *Shimoni bat lyssavirus* (SHIBV), and phylogroup III is represented by *Ikoma lyssavirus* (IKOV), *Lleida bat lyssavirus* (LLBV) and *West Caucasian bat lyssavirus* (WCBV) [4,6,7]. The rabies virus (RABV) is distributed worldwide, except for Japan, Ireland, Great Britain, New Zealand, Hawaii, Australia, Antarctica, and several Western European countries [4,8]. In the Americas, RABV is the only lyssavirus found [9]. Rabies virus circulates through two different epidemiological cycles: terrestrial and bat-borne, associated with different species within the orders *Carnivora* and *Chiroptera*, respectively. In Argentina, two terrestrial variants are found: V1 associated with domestic dogs; V2 circulates among sylvatic canids, particularly in the crab-eating fox, *Cerdocyon thous* [10,11]. Argentinian bat-borne variants can be subdivided into two groups: those circulating among insectivorous bats and variants associated with the hematophagous bat *Desmodus rotundus*. Five genus-specific distinct lineages were described in insectivorous bats associated with *Tadarida*, *Myotis*, *Eptesicus*, *Histiotus*, and *Lasiurus-Dasypterus* species from Argentina [12]. Although the prevalence of insectivorous bat-RABV is conspicuous, no human cases have yet been associated with these variants. The common vampire bat, *Desmodus rotundus*, is the main rabies sylvatic reservoir in Argentina, with high prevalence levels in northern provinces [13]. It is an enzootic disease in bovines from this region but also occurs in epizootic waves from distant territories, affecting thousands of cattle every year, and it was also responsible for two human deaths in 1997 and 2001 [14]. Two main groups of vampire bat-transmitted RABV have been recognized through phylogenetic analysis of the nucleoprotein gene sequence: V3 and V3A, distributed in the northeast (NEA) and the northwest (NWA) of Argentina, respectively [15].

Given the high lethality of rabies disease, in epidemiological surveillance, it is essential to count on a rapid diagnosis to give a quick response. The direct fluorescent antibody test (FAT) is the most frequently used method for the diagnosis of rabies, because of its high sensitivity and specificity [16,17]. During the past two decades, various conventional RT-PCR protocols were developed for the diagnostic amplification of RABV [16,17,18]. These assays proved to be both sensitive and specific tools for routine diagnosis, especially in decomposed samples [19,20]. Posteriorly, the introduction of fluorogenic probes enabled specific diagnosis by a hybridization reaction. Several Real Time RT-PCR (qRT-PCR) protocols were developed using the TaqMan technology, which allows for rapid detection of RABV [16,17,18]. However, these probes and primers are often designed based on a limited number of sequences and thus may not represent the genetic diversity of all current variants. For example, single mutations for North American RABV strains in the region of the primers or the probe have been shown to alter the sensitivity of the qRT-PCR [21]. Thus, validation of probe-based assays in a diagnostics laboratory should be conducted to confirm the detection of all circulating variants in a specific region.

Molecular techniques (including TaqMan assays) have been considered as reference methods by the OIE for the post-mortem diagnosis of animal rabies [17]. A widely used TaqMan RT-qPCR assay, especially in European national reference laboratories [22] is based on the universal pan-lyssavirus primers JW12 and N165-146 and the LysGT1 TaqMan RABV probe developed by Wakeley et al. [23]. This oligo set spans a 111 nucleotide region including the transcription initiation signal and part of the coding sequence of the nucleoprotein (N) gene. This set was designed for the specific detection of classical RABV and showed high levels of specificity and sensitivity [22,23,24]. More recently, the LN34 primers and probes set—another increasingly implemented assay—was designed and validated [25]. This assay amplifies a 165 nucleotide region including the leader sequence, the transcription initiation signal, and part of the coding sequence of the N gene, being partially overlapped with the locus targeted by the LysGT1 assay. The LN34 assay was validated using a panel containing representative members of major worldwide RABV variants and 13 other lyssaviruses, employing multiplex, degenerate primers. It incorporates a degenerate, locked nucleotide (LNA) modified probe, which confers higher oligo stability and increases sequence melting temperature (Tm), allowing for the design of shorter probes which in turn are more effectively quenched and have a higher signal-to-noise ratio. Consequently, these are more sensitive probes, which provide robust target detection [26].

In this study, we test the applicability of these two widely used primers/probes sets in all variants circulating in Argentina, with special emphasis on vampire-bat transmitted RABV genetic variability. All variants were successfully detected using the LN34 assay, while the LysGT1 assay is prone to yield variant-dependent false-negative test results. Therefore, the LN34 assay would ensure more effective detection of RABV in Argentina.

## 2. Materials and Methods

### 2.1. Samples

A total of 44 samples obtained from bats, dogs, cattle, and horses that were previously tested for rabies by conventional RT-PCR and FAT, were analyzed by qRT-PCR LysGT1 and LN34 assays. All samples were received at the Instituto de Zoonosis Luis Pasteur in the period 2018–2020. The year of submission and place of origin are shown in Table 1 and Table 2 and Figure 1. For FAT, Ammon’s horns, brainstem, and cerebellum, or the whole brain, for dogs, cattle and horses, and bats, respectively, were used following the OIE protocols [17]. A 10% tissue homogenate of the mentioned brain structures of each specimen was prepared in Minimum Essential Medium (MEM, Thermo Fisher Scientific, Waltham, MA, USA) following the World Health Organization (WHO) guidelines [27].

### 2.2. FAT

Slides and controls were stained with an antinucleocapsid antibody (BIO-RAD, Marne-La-Coquette, France) according to the manufacturer’s instructions.

### 2.3. Viral RNA Extraction

Viral nucleic acid extraction was performed using the High Pure Viral RNA kit (Roche Molecular Biochemicals, Mannheim, Germany) according to the manufacturers’ instructions contained in the kit insert.

### 2.4. RT-PCR and DNA Sequencing

Reverse transcription and end-point PCR amplification were performed using primers 504 (Sense: 5′ TATACTCGAATCATGATGAATGGAGGTCGACT 3′) and 304 (Antisense: 5′ TTGACGAAGATCTTGCTCAT 3′) using the OneStep RT-PCR Kit (Qiagen, Venlo, The Netherlands) under conditions described by Orciari et al. [28]. Amplification products were checked by 2% agarose gel electrophoresis stained with ethidium bromide. A booster PCR was performed using the same primer pair. The product was quantified using agarose gel electrophoresis and sequenced using a BigDye Terminator v3.1 cycle sequencing kit according to the manufacturer’s protocol using the ABI PRISM 310 Genetic Analyzer (Applied Biosystems Inc. Foster City, CA, USA).

### 2.5. Phylogenetic Analysis

A 191-bp region, obtained with primers 504 and 304 was analyzed (Appendix A). In addition to the samples that were subjected to qRT-PCR and sequenced, we sequenced 11 samples typed as V3 (6 samples) and V3A (5 samples) by Torres et al. [15]. We included 32 additional reference sequences corresponding to all RABV variants circulating in Argentina, retrieved from Genbank. Nucleotide alignments were performed with Clustal Omega [29].

The phylogenetic analysis was performed in MrBayes 3.2.7a [30] on the CIPRES Science Gateway [31]. Nucleotide substitution models were estimated using MrModeltest [32] under the Akaike Information Criterion (correcting by the number of taxa). The selected substitution model was K80.

We ran Markov chain Monte Carlo (MCMC) analyses for two independent runs for 5 × 10^7^ generations, sampling every 5 × 10^3^. Convergence was assessed by analyzing the potential scale reduction factor (PSRF) and the average standard deviation of split frequencies (ASDSF). The “burnin” phase was set up in the generation which fulfilled PSRF values of 1.00–1.02 for all estimated parameters and standard deviations lower than 0.01, which corresponded to 9.49% of the total run. Trees were visualized with iTOL [33] and Figtree [34].

### 2.6. LysGT1/β-Actin qRT-PCR

The LysGT1 qRT-PCR was performed in a duplex reaction with the β-actin gene, following Wakeley et al. [23], except for the following modifications. Duplex qRT-PCR was performed using a total volume of 20 µL on 5 µl of extracted RNA and 15 µl of the master mix components from the SensiFAST™ Probe No-ROX One-Step Kit (Bioline, London, UK), following manufacturer’s instructions. The LysGT1/β-actin qRT-PCR was performed on an Applied Biosystems 7500 Real-Time PCR System (Applied Biosystems, Foster City, CA, USA). The following thermal cycling conditions were used: 48 °C for 15 min (RT), 95 °C for 10 min (inactivation), followed by 45 cycles of 95 °C for 30 s, 52 °C for 30 s, and 72 °C for 20 s. Critical threshold cycle numbers (Ct) were determined automatically with the Applied 7500 Real-Time PCR System (Applied Biosystems, Foster City, CA, USA). Negative (FAT and RT-PCR negative samples) and positive controls (CVS RNA) were included in each assay for its validation.

### 2.7. LN34 qRT-PCR

The LN34 assay was carried out following Wadhwa et al. [25]. Samples were assayed in a 25-µL reaction mixture containing 2 µL of RNA, with the AgPath-ID™ One-Step RT-PCR kit (Life Technologies, Carlsbad, CA, USA). The cycling conditions were as follows: reverse transcription at 50 °C for 30 min, followed by RT inactivation/initial denaturation at 95 °C for 10 min, and amplification for 45 cycles at 95 °C for 15 s and 56 °C for 30 s on an Applied Biosystems 7500 Real-Time PCR System (Applied Biosystems, Foster City, CA, USA). Corresponding controls were run in each assay. A sample was considered positive if the LN34 Ct ≤ 35.

### 2.8. LysGT1 qRT-PCR Amplicon Sequencing and Alignment

A 111 bp region of the nucleoprotein gene comprising the amplicon used in qRT-PCR was sequenced in 32 samples covering all variants circulating in Argentina, performing RT-PCR and a subsequent sequencing protocol using primers 001-921B and 550F as described previously [15]. Sequences were deposited in Genbank (Accession Numbers: MW302176-MW302207). The 32 sequences were aligned with a reference RABV genome isolated from *Desmodus rotundus* (Genbank Accession Number: KU523255). Real Time RT-PCR primers JW12 and N165-146 as well as the probe LysGT1 were mapped in the resulting alignment. Appendix A shows a schematic representation of the probes and the four total primer pairs used in this study.

## 3. Results

A total of 44 samples corresponding to bats, dogs, cattle, and horses, with a positive diagnosis by conventional RT-PCR and/or FAT, were tested by qRT-PCR using LN34 and LysGT1 assays. All variants were successfully detected using the LN34 primers and probe set. In contrast, the LysGT1 assay positively detected variants V1, V2, V3, V4, Eptesicus, and Histiotus but failed to detect variants V3A, V6, and Myotis (Table 1 and Table 2). Notably, there was a relatively high level of discrepancy between FAT and RT-PCR: 7 samples out of 44 were negative by FAT while positive by RT-PCR (Table 1 and Table 2).

The molecular typing based on 191 bp involving the 3′ terminus of the N gene and 51 downstream nucleotides was unequivocal. The final matrix contained 87 sequences (Appendix A). No internal gaps or stop codons were detected in the alignment. The Bayesian phylogeny recovered the expected relationships among variants circulating in Argentina (Figure 2). Namely, the two epidemiological cycles (terrestrial and bat-borne) were reciprocally monophyletic, as well as each particular variant within them, confirming that the employed primer set produces informative sequences for phylogenetic genotyping. Both vampire bat-transmitted subgroups, V3 and V3A, were also reciprocally monophyletic, showing that they constitute well-differentiated RABV lineages.

The 19 samples typed as V3A (Figure 2) were negative by the LysGT1 assay (Table 1). In contrast, the seven samples typed as V3 were positive by this assay (Figure 2, Table 1). The region comprising both primers is invariant among V3 and V3A (Figure 3); consequently, the difference in RABV detection by qRT-PCR lies in the number of mismatches in the probe region. Variant V3 has a G instead of A in the base number 17 of the LysGT1 probe sequence. This substitution is also present in V3A, which has two additional mismatches: a G (instead of A) in base 11 and a C (instead of T) in base 23, in all sequences (Figure 3).

As mentioned, all V3 and V3A samples yielded detectable products in the LN34 assay, although sample 973 (V3A) had a Ct >35 which should be considered inconclusive and repeated according to the current LN34 protocol [35]. Indeed, we repeated three times the RNA extraction and subsequent qPCR protocol and obtained Ct values between 37 and 38. Notably, the β-actin Cts were repeatedly near 25, and it is undoubtedly a positive sample since we obtained FAT and RT-PCR positive results (Table 1), and we further successfully sequenced the 249 bp amplicon enabling variant determination (Figure 2). We interpret that a Ct value of 37, although inconclusive per se, falls within the limit of detection of the LN34 assay.

Twelve out of 18 samples typed as terrestrial and non-vampire-bat variants were positive by the LysGT1 assay (Figure 2, Table 2). Terrestrial variants V1 and V2 were detected in all cases and depict only one mismatch with respect to the probe sequence (Figure 3, Table 2). The most frequent variant circulating in Argentina is V4 which belongs to the widespread bat *Tadarida brasiliensis*. This variant was successfully amplified and detected by qRT-PCR in the four individuals analyzed, one of them showing two mismatches in the region comprising primer N165-146, while the remaining having one mismatch (Figure 3, Table 2). The variant circulating among bats of the genus *Histiotus* was also positively diagnosed by this method, both samples depicting 1 mismatch in the region of primer N165-146. In contrast, the variant circulating in *Lasiurus* and *Dasypterus* (V6) was negatively diagnosed. The two samples showed three mismatches: two in the probe region and one in the primer N165-146. The three Myotis samples were also undetected and depicted the highest levels of variation in the region delimited by primers JW12 and N165-146, showing two and three mismatches in the probe region and two mismatches in the region corresponding to primer N165-146. One of the two samples typed as Eptesicus was positive by qRT-PCR, while the other did not produce detectable amplification by this method. We could only obtain a good quality sequence of the region encompassing the qRT-PCR oligos in the positively diagnosed sample, which showed two total mismatches, one in the probe region and one with respect to primer N165-146.

All terrestrial and non-vampire-bat variants were detected by the LN34 assay and considered positive (Ct ≤ 35) according to the current protocol [35]. The only exception was sample 175, typed as Eptesicus, which yielded a Ct of 39 (Table 2). We repeated three times the RNA extraction and obtained Ct values of 39. Again, the β-actin Ct values were low (between 23 and 24). Although it is undoubtedly a positive sample since we obtained RT-PCR positive results (Table 1) and sequenced the 249 bp amplicon (Figure 2), it was negative by FAT and, as mentioned, by the LysGT1 assay. As occurred with sample 973 (V3A), we interpret that a Ct ≤35, although inconclusive, is (weakly) detectable by the LN34 assay.

## 4. Discussion

The World Organization for Animal Health (OIE) recommends testing diagnostic assay performance for virus variants circulating in a certain working area [16]. In Argentina, there are at least nine different RABV genetic lineages. Two variants are restricted to canids, five of them are found in specific insectivorous bat genera and two related variants circulate among vampire bats. Validation of a RABV qRT-PCR protocol in Argentina should accomplish the correct amplification of all circulating variants, especially those with a higher incidence in public health.

Crucial for this survey was the unequivocal RABV variant identification. Previous studies have shown that complete [15] or partial [10,12] N gene sequences successfully discriminated variants circulating in Argentina. However, partial sequence analyses were obtained from amplicons using primers 10 g and 304, which are 1449 bp long [36]. In this study, we show that the combination of primers 504 and 304 is capable of separating all circulant RABV variants from Argentina (Figure 2). This is an important advantage for laboratories involved in epidemiological surveillance since this primer set produces a short amplicon (249 bp), which has been shown to be more sensitive especially in decomposed samples [20], which would account for the relatively high discrepancies (16%) between FAT and RT-PCR results (Table 1 and Table 2).

In this study, we compared the effectiveness of two widely used sets of qRT-PCR primers and probes spanning partially overlapping regions of the nucleoprotein gene for the detection of classical rabies virus, in all variants circulating in Argentina. Both assays, LysGT1 and LN34, were designed based on a high number of lyssavirus variants, although the former was specifically designed for RABV detection, while the latter is a pan-lyssavirus assay [23,25]. Both assays have shown comparable detection limits: 16.7–25.8 copies/µL for LysGT1 [22] and eight copies (95% confidence interval: 0–18 copies) for LN34 [26]. Hence both were expected to positively detect RABV variants circulating in Argentina.

The LysGT1 assay proved to be successful in RABV detection of terrestrial V1 and V2, as well as insectivorous-related variants V4 and Histiotus, all of which depict less than two total mismatches with respect to the region encompassing both primers (JW12 and N165-146) and the probe (LysGT1) (Figure 3, Table 2). However, it failed in the positive detection of RABV variants of insectivorous bats of the genera *Myotis* and *Lasiurus*-*Dasypterus* (Figure 3, Table 2). A simple rule can be established to predict the performance of the qRT-PCR carried out with the LysGT1 assay: a maximum of two total mismatches with respect to the primers and probe sequences allows RABV detection, while the presence of three or more total mismatches precludes viral detection with no exception. Given that mismatches in the probe sequences have more impact on both sensitivity and specificity of TaqMan qRT-PCR than mismatches in primer sequences [37,38], another nonexclusive rule could be established if only the probe is taken into consideration: a maximum of one mismatch allows viral detection, while two or more substitutions hinder positive diagnosis. Both rules apply for all variants, independent of their origin (terrestrial, insectivorous bat-related, or vampire bat-related variants). The only variant for which we obtained contrasting results was Eptesicus (Table 2). It is an infrequent variant and we could only include two samples, one of which was positively detected and fits the above-mentioned rule, since it shows a mismatch in the probe region and another in primer N165-146, totalizing two mismatches. We speculate that the other sample had at least an extra mismatch, but we could not obtain a good quality sequence of this region.

As mentioned above, in Argentina, vampire transmitted-RABV produces thousands of deaths in cattle yearly and has caused two human deaths in the past 25 years [14]. Although infecting the same host species, V3 and V3A have been shown to be genetically distinct in analyses based on the nucleoprotein gene (N) nucleotide sequences [12]], this study. Indeed, these two variants differ in the region encompassing the probe LysGT1, which is determinant for the performance of this qRT-PCR assay.

All variants were successfully detected using the LN34 assay, making it the preferable alternative compared with LysGT1. The only two samples that were inconclusive according to the developers’ protocol [35] yielded detectable products but with high (≤35), but reproducible, Ct values. The LN34 assay was validated in a multisite study involving 14 laboratories, testing almost 3000 samples from the Americas, Africa, Asia, Europe, and the Middle East [26]. This assay showed the capability of detecting fresh tissue samples but also frozen, deteriorated, and even formalin-fixed brain tissue. The LN34 depicted high specificity (99.68%) and sensitivity (99.90%) compared with the FAT test and enabled the identification of both FAT false-positive and false-negative results [26]. As occurs with the LysGT1 assay in the present study, the LN34 assay has a tolerance of a maximum of 1 mismatch in the probe sequences [26]. In a recent study based on formalin-fixed human brain tissue, the LN34 assay proved to be more sensitive than traditional PCR, probably due to the use of a shorter amplicon [39]. Furthermore, direct sequencing of the 165 bp LN34 amplicon allowed rapid and low-cost sample genetic typing. Taking into account the above-mentioned evidence, and in addition to the results presented in this study, the LN34 assay can be considered a reliable and robust method for rabies diagnostics and typing of variants circulating in Argentina and, at least, in the Southern Cone of South America.

## 5. Conclusions

In the present study, we showed the extent and limits of two TaqMan qRT-PCR assays for RABV diagnosis in Argentina. We conclude that the LysGT1 assay is prone to yield variant-dependent false-negative test results, and in consequence, the use of LN34 assay in diagnostic laboratories would ensure more effective detection of circulating RABV variants in Argentina.

## Figures and Tables

**Figure 1 viruses-13-00023-f001:**
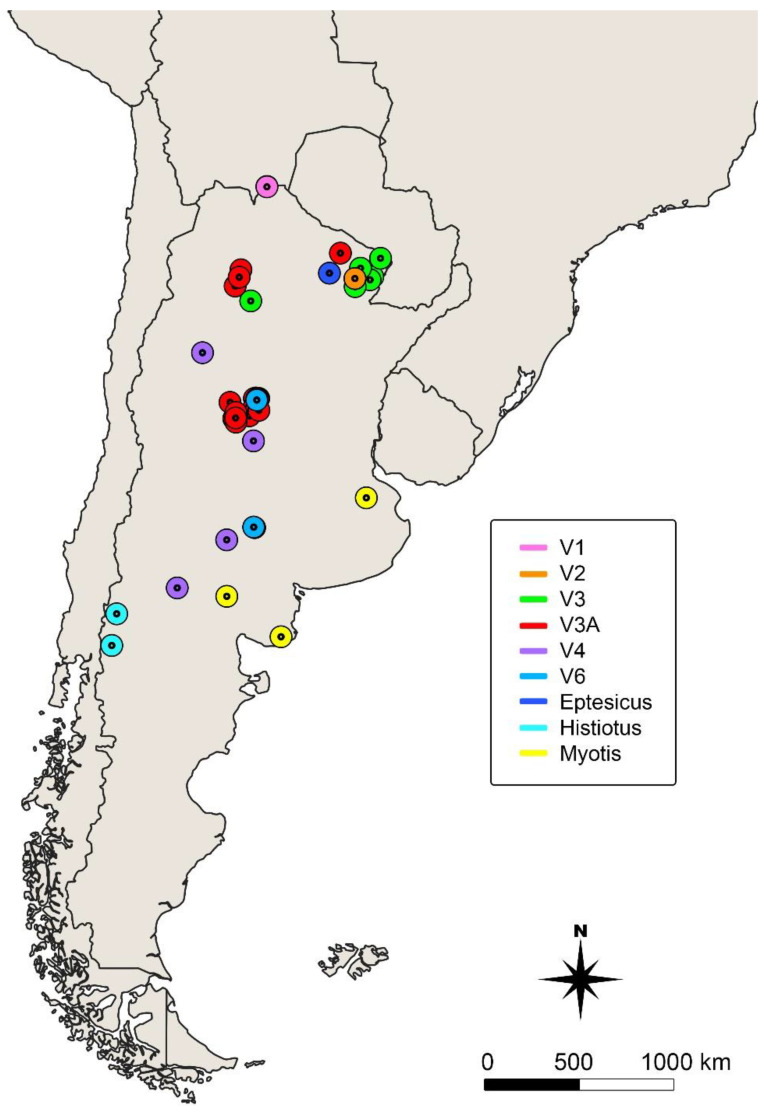
Map of Argentina, showing sampling localities. Colors indicate rabies virus (RABV) genetic variants.

**Figure 2 viruses-13-00023-f002:**
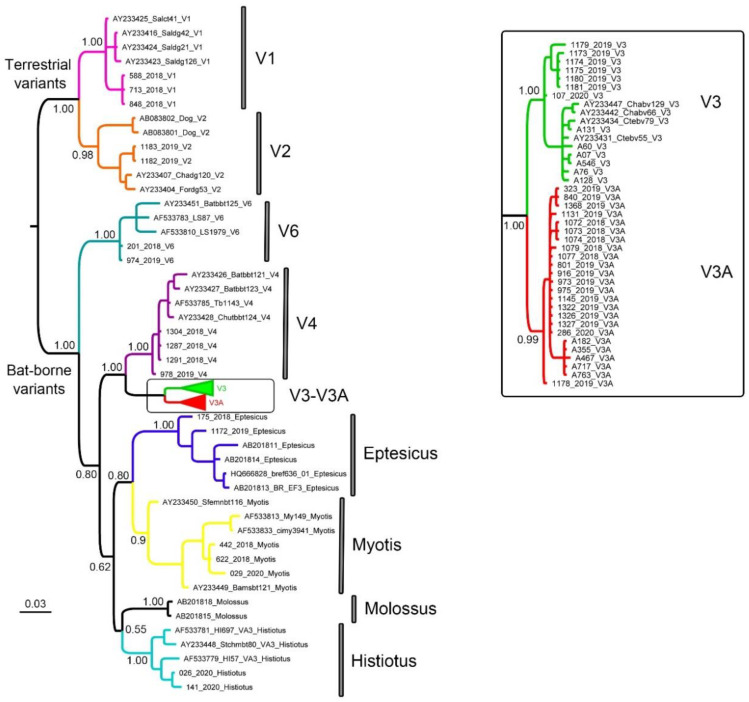
Bayesian phylogeny of 87 sequences based on 191 bp involving the 3′ terminus of the RABV N gene and 51 downstream nucleotides. Bayesian posterior probability of main clades is shown, as well as the scale (substitutions/site). In the left panel, the subclade (V3, V3A) is expanded.

**Figure 3 viruses-13-00023-f003:**
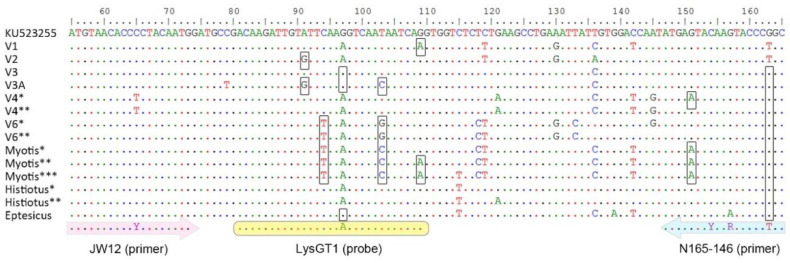
Alignment showing variability of Argentinian RABV genetic variants in the region comprising the amplicon used in the LysGT1 qRT-PCR assay. Primers JW12 and N165-146 and the probe LysGT1 were mapped in the resulting alignment. Dots depict conserved nucleotides. Mismatches of the primers and probe are indicated with black empty boxes. The sequence used as reference is a variant isolated from *Desmodus rotundus* (Genbank Accession Number: KU523255). The number of asterisks after variant denomination corresponds to different samples (Table 2). The remaining RABV variants depict no variation across samples in the region shown.

**Table 1 viruses-13-00023-t001:** Samples characterized as V3 and V3A analyzed in this study. Source, year of submission, place of origin, Direct Fluorescent Antibody Test (FAT), RT-PCR, and qRT-PCR results are shown. The number of mismatches in the primers/probe region of the LysGT1 oligos is shown (right).

									qRT-PCR (Ct)				Mismatches			
Sample ID	Year	Organism	Province	Locality	Latitude	Longitude	FAT	RT-PCR	LN34	LysGT1	β-actin	Variant	JW12	LysGT1	N165-146	Total
1072	2018	Horse	Córdoba	La Calera	−31.35	−64.34	Negative	Positive	28	-	25	V3A	NS			-
1073	2018	Horse	Córdoba	La Calera	−31.35	−64.34	Negative	Positive	29	-	24	V3A	NS			-
1074	2018	Vampire bat	Córdoba	Unknown (Colón) *	-	-	Negative	Positive	18	-	25	V3A	NS			-
1077	2018	Horse	Córdoba	La Calera	−31.35	−64.34	Negative	Positive	22	-	22	V3A	NS			-
1079	2018	Bovine	Córdoba	Ambul	−31.49	−65.06	Positive	Positive	24	-	29	V3A	NS			-
323	2019	Bovine	Tucumán	Ticucho	−26.52	−65.25	Positive	Positive	26	-	25	V3A	0	3	1	4
801	2019	Bovine	Córdoba	Santa María	−31.68	−64.31	Positive	Positive	27	-	24	V3A	0	3	1	4
840	2019	Bovine	Salta	Rosario de la Frontera	−25.80	−64.97	Positive	Positive	26	-	25	V3A	0	3	1	4
916	2019	Bovine	Córdoba	Calamuchita	−32.08	−64.55	Positive	Positive	26	-	21	V3A	0	3	1	4
973	2019	Horse	Córdoba	Santa María	−31.72	−64.18	Positive	Positive	37	-	25	V3A	NS			-
975	2019	Bovine	Córdoba	Santa María	−31.85	−64.08	Positive	Positive	27	-	27	V3A	0	3	1	4
1131	2019	Bovine	San Luis	Junín	−32.20	−65.32	Positive	Positive	24	-	22	V3A	NS			-
1145	2019	Bovine	Córdoba	Córdoba Capital	−31.40	−64.22	Positive	Positive	24	-	23	V3A	0	3	1	4
1178	2019	Bovine	Formosa	Estanislao del Campo	−25.06	−60.08	Positive	Positive	20	-	25	V3A	0	3	1	4
1322	2019	Bovine	San Luis	Santa Rosa de Conlara	−32.35	−65.21	Positive	Positive	27	-	25	V3A	NS			-
1326	2019	Bovine	Córdoba	La Patria	−31.52	−65.50	Positive	Positive	32	-	26	V3A	NS			-
1327	2019	Bovine	Córdoba	Villa Dolores	−31.94	−65.21	Positive	Positive	28	-	28	V3A	0	3	1	4
1368	2019	Bovine	Salta	La Candelaria	−26.13	−65.05	Positive	Positive	28	-	26	V3A	0	3	1	4
286	2020	Bovine	Córdoba	Tilquicho	−32.17	−65.23	Positive	Positive	23	-	24	V3A	0	3	1	4
1173	2019	Bovine	Formosa	Mariano Boedo	−26.11	−58.49	Positive	Positive	15	13	21	V3	0	1	1	2
1174	2019	Bovine	Formosa	Misión Laishi	−26.24	−58.62	Positive	Positive	17	13	20	V3	0	1	1	2
1175	2019	Bovine	Formosa	Pirané	−25.73	−59.09	Positive	Positive	18	14	20	V3	0	1	1	2
1179	2019	Bovine	Formosa	Salvación	−25.29	−58.11	Positive	Positive	16	12	20	V3	0	1	1	2
1180	2019	Bovine	Chaco	Gral. San Martín	−26.54	−59.36	Positive	Positive	17	14	23	V3	0	1	1	2
1181	2019	Bovine	Formosa	El Salado	−25.15	−59.39	Positive	Positive	24	18	23	V3	0	1	1	2
107	2020	Bovine	Santiago del Estero	Pozo Hondo	−27.17	−64.49	Positive	Positive	29	22	24	V3	0	1	1	2

* The locality of sample 1074 is unknown but belongs to the Department of Colón (Córdoba). NS: Samples that were not sequenced in the LysGT1 qRT-PCR oligos region.

**Table 2 viruses-13-00023-t002:** Samples characterized as V1, V2, V4, V6, Myotis, Eptesicus, and Histiotus, analyzed in this study. Source, year of submission, place of origin, Direct Fluorescent Antibody Test (FAT), RT-PCR, and qRT-PCR results are shown. The number of asterisks indicates the denomination of each sequence in Figure 3. The number of mismatches in the primers/probe region of the LysGT1 oligos is shown (right).

									qRT-PCR (Ct)				Mismatches			
Sample ID	Year	Organism	Province	Locality	Latitude	Longitude	FAT	RT-PCR	LN34	LysGT1	β-actin	Variant	JW12	LysGT1	N165-146	Total
588	2018	Dog	Salta	Salvador Mazza	−22.05	−63.69	Negative	Positive	25	19	25	V1	0	1	0	1
713	2018	Dog	Salta	Salvador Mazza	−22.05	−63.69	Positive	Positive	26	22	21	V1	NA			-
848	2018	Dog	Salta	Salvador Mazza	−22.07	−63.69	Positive	Positive	22	18	23	V1	0	1	0	1
1182	2019	Dog	Formosa	Colonia El Bañadero	−26.19	−59.36	Positive	Positive	22	20	23	V2	NA			-
1183	2019	Dog	Formosa	Colonia El Bañadero	−26.18	−59.37	Positive	Positive	22	20	29	V2	0	1	0	1
1287	2018	Bat	Neuquén	Neuquén	−38.97	−68.10	Positive	Positive	21	21	18	V4 *	0	0	2	2
1291	2018	Bat	Córdoba	Río Cuarto	−33.12	−64.35	Positive	Positive	21	15	24	V4 **	0	0	1	1
1304	2018	Bat	La Pampa	Santa Rosa	−37.10	−65.67	Positive	Positive	19	14	21	V4 **	0	0	1	1
978	2019	Bat	La Rioja	La Rioja	−29.41	−66.85	Positive	Positive	22	19	22	V4 **	0	0	1	1
201	2018	Bat	La Pampa	Santa Rosa	−36.60	−64.34	Negative	Positive	25	-	21	V6 *	0	2	1	3
974	2019	Bat	Córdoba	Córdoba Capital	−31.42	−64.19	Positive	Positive	27	-	24	V6 **	0	2	1	3
442	2018	Bat	Buenos Aires	San Miguel del Monte	−35.44	−58.81	Positive	Positive	22	-	20	Myotis ***	0	2	2	4
622	2018	Bat	Río Negro	Viedma	−40.81	−63.00	Positive	Positive	21	-	22	Myotis ****	0	3	2	5
29	2020	Bat	Río Negro	Choele Choel	−39.29	−65.66	Positive	Positive	20	-	20	Myotis *****	0	3	2	5
26	2020	Bat	Neuquén	Junín de los Andes	−39.95	−71.07	Positive	Positive	24	18	26	Histiotus *	0	0	1	1
141	2020	Bat	Río Negro	Bariloche	−41.13	−71.31	Positive	Positive	22	17	19	Histiotus **	0	0	1	1
175	2018	Bat	La Pampa	Santa Rosa	−36.62	−64.29	Negative	Positive	39	-	23	Eptesicus	NA			-
1172	2019	Bat	Chaco	Castelli	−25.95	−60.62	Positive	Positive	24	19	23	Eptesicus	0	1	1	2

The number of asterisks indicates the denomination of each sequence in Figure 3. NA: Not applicable. Samples for which fresh material was consumed with no possibility of obtaining good quality nucleotide sequences of the region encompassing LysGT1 qRT-PCR primers and probe.

## Data Availability

Data is available in the article and supplementary material.

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
