# Peer review of "Evaluation of Two Real-Time, TaqMan Reverse Transcription-PCR Assays for Detection of Rabies Virus in Circulating Variants from Argentina: Influence of Sequence Variation"

_viruses, 2020, doi:10.3390/v13010023_

Round 1

Reviewer 1 Report

Dear authors, 

This publication gives an overview of the potential limitation of pan-lyssavirus assays for the detection of RABV species in specific locations. 

Considering the introduction: Lyssaviruses are no longer subdivided into genotypes but are rather referred to as Lyssavirus species. For example, Rabies virus is no longer referred to as Lyssavirus genotype 1 but as RABV. 
In addition, in the mean time at least 16 different species have been described in contrast to the 7 that are mentioned in your introduction. Please update this part of the manuscript. 

Author Response

Response to reviewers

General considerations: we have used the "Track Changes" function to make the modifications visible to Reviewers and the Editor. We detected minor typing errors in the text, which have all been corrected. Figure 3 was updated ("Molosus" was changed to "Molossus") (Line 230 of the final version). As new citations were included and some previous citations were removed, the reference list was updated (Lines 370-481 of the final version).

Reviewer 1

Considering the introduction: Lyssaviruses are no longer subdivided into genotypes but are rather referred to as Lyssavirus species. For example, Rabies virus is no longer referred to as Lyssavirus genotype 1 but as RABV. 
In addition, in the mean time at least 16 different species have been described in contrast to the 7 that are mentioned in your introduction. Please update this part of the manuscript. 

Response: We now refer to the current classification of Lyssaviruses, which recognizes 17 species, following the International Committee on Taxonomy of Viruses (www.ictvonline.org/) and supporting bibliography (Lines 39-51 of the final version). We refer to species rather than genotypes throughout the paper.

Reviewer 2 Report

As the authors described that the LN34 assay is the most appropriate detection method for circulating RABV genome  not only classical dogs/wild animals rabies but also rabies species of bats. Of course, I also agree the credibility of LN34. Is there any advantage of use of LysGT1 assay? Do you have any comments? If you feel any merits, please clarify this point in the discussion section.

Minor comments:

1) Lines 41-43: Japan is also rabies-free country over sixty years long.

2) Do the authors have any implication of making Table 1 and 2, separately? Total 44 samples' data can be combined on one Table.

3) Table 1: Characters of table should be aligned. eg; LN3, 4 should be LN34.

Author Response

Response to reviewers

General considerations: we have used the "Track Changes" function to make the modifications visible to Reviewers and the Editor. We detected minor typing errors in the text, which have all been corrected. Figure 3 was updated ("Molosus" was changed to "Molossus") (Line 230 of the final version). As new citations were included and some previous citations were removed, the reference list was updated (Lines 370-481 of the final version).

Reviewer 2

As the authors described that the LN34 assay is the most appropriate detection method for circulating RABV genome  not only classical dogs/wild animals rabies but also rabies species of bats. Of course, I also agree the credibility of LN34. Is there any advantage of use of LysGT1 assay? Do you have any comments? If you feel any merits, please clarify this point in the discussion section.

Response: According to our results, we have no reason to support the use of the LysGT1 assay, at least in the region under study. Nevertheless, we cannot extrapolate our interpretations beyond the samples and variants that were studied. As we comment in the Introduction the LysGt1 assay was developed before the LN34 assay and widely adopted by diagnostic laboratories. However it has been gradually replaced after the validation of the LN34 assay. Our study confirms that the LN34 outperforms the results obtained with LysGT1 in variants from Argentina, emphasizing the need to adopt the former assay in diagnostic laboratories in our region.

Minor comments:

1) Lines 41-43: Japan is also rabies-free country over sixty years long (Line 52 of the final version).

Response: The omission was corrected. Thank you for pointing it out.

2) Do the authors have any implication of making Table 1 and 2, separately? Total 44 samples' data can be combined on one Table.

Response: We consider that keeping two separate tables would be easier for the reader. We have incorporated a high number of vampire bat samples, which include two subvariants that yielded contrasting results, and we think that showing them as a separate table enables a rapid distinction of the differences among both subvariants. Besides, the unification of both tables would produce an extremely large one.

3) Table 1: Characters of table should be aligned. eg; LN3, 4 should be LN34.

Response: Modified as requested (Line 196 of the final version).